# Adhesion Molecule Targeted Therapy for Non-Infectious Uveitis

**DOI:** 10.3390/ijms23010503

**Published:** 2022-01-03

**Authors:** Yi-Hsing Chen, Sue Lightman, Malihe Eskandarpour, Virginia L. Calder

**Affiliations:** 1UCL Institute of Ophthalmology, University College London, London EC1V 9EL, UK; yi-hsing.chen.14@alumni.ucl.ac.uk (Y.-H.C.); s.lightman@ucl.ac.uk (S.L.); m.eskandarpour@ucl.ac.uk (M.E.); 2Department of Ophthalmology, Chang Gung Memorial Hospital, Taoyuan 333, Taiwan; 3College of Medicine, Chang Gung University, Taoyuan 333, Taiwan; 4NIHR Biomedical Research Centre at Moorfields Eye Hospital NHS Foundation Trust, London EC1V 2PD, UK

**Keywords:** adhesion molecule, integrin, lymphocyte function-associated antigen-1 (LFA-1), very late antigen-4 (VLA-4), uveitis, experimental autoimmune uveitis, non-infectious uveitis, intercellular cell adhesion molecule-1 (ICAM-1), vascular cell adhesion protein 1 (VCAM-1), selectin

## Abstract

Non-infectious uveitis (NIU) is an inflammatory eye disease initiated via CD4^+^ T-cell activation and transmigration, resulting in focal retinal tissue damage and visual acuity disturbance. Cell adhesion molecules (CAMs) are activated during the inflammatory process to facilitate the leukocyte recruitment cascade. Our review focused on CAM-targeted therapies in experimental autoimmune uveitis (EAU) and NIU. We concluded that CAM-based therapies have demonstrated benefits for controlling EAU severity with decreases in immune cell migration, especially via ICAM-1/LFA-1 and VCAM-1/VLA-4 (integrin) pathways. P-selectin and E-selectin are more involved specifically in uveitis related to vasculitis. These therapies have potential clinical applications for the development of a more personalized and specific treatment. Localized therapies are the future direction to avoid serious systemic side effects.

## 1. Introduction

Non-infectious uveitis (NIU) describes a broad disease spectrum involving inflammation in the pigmented middle layer of the eye (uvea) driven by autoimmune and autoinflammatory pathways [1,2]. Epidemiologically, NIU has a higher prevalence in people of working age, and a significant socioeconomical impact [3,4]. Pathologically, this intraocular inflammation is mediated via antigen-specific CD4^+^ T cells becoming activated, which then proliferate centrally and migrate into the eye. As a result of this, inflammatory myeloid cells are recruited and contribute to the retinal structural damage and final vision disturbance [5].

Recruitment of activated effector CD4^+^ T cells to the immune reaction site involves a well-defined cascade, beginning with capturing freely moving leukocytes to the endothelium, which roll before adhering to endothelial cells, undergoing post-adhesion strengthening, being activated, crawling, and transmigrating across the endothelium. During the steps of the leukocyte recruitment cascade, a distinct set of adhesion molecules is activated to fine-tune the process temporally and spatially [6,7]. Cell adhesion molecules (CAM) are glycoproteins expressed on the cell surface and include the integrin family, the immunoglobulin superfamily, selectins, and cadherins [8]. In this review, it was noticed that the involvement of integrins, selectins, and cadherins was mainly reported in uveitis [9]. Integrin assembly constitutes obligate noncovalently bound heterodimers. In mammals, subunits of 18 α and 8 β integrins within the integrin family have been identified to form 24 different integrin heterodimers [10]. Among the leukocyte subsets, it is known that neutrophils express mostly β2-integrins with a few β1- and β3-integrins. Monocytes mainly express β1- and β2-integrins, whereas lymphocytes predominantly express β1-, β2-, and β7-integrins varying by the state of activation and their subtype. Specifically, leukocytes express heterodimeric integrins including αLβ2 (lymphocyte function-associated antigen-1 (LFA-1), CD11a/CD18), αMβ2 (macrophage antigen-1 (Mac-1), CD11b/CD18), αXβ2, αDβ2, α4β1 (very late antigen-4 (VLA-4), CD49d/CD29), and α4β7. Along with other adhesion molecules, such as selectin, intercellular cell adhesion molecule-1 (ICAM-1), and vascular cell adhesion protein 1 (VCAM-1), these molecules govern immune cell migration to sites of inflammation and T-cell-mediated immunity in tissues [6].

The importance of CAMs involved differentially in the pathways of many autoimmune diseases has been validated by selectively targeting integrins for clinical use and monitoring their clinical efficacy. For example, in another immune-mediated central nervous system disease, multiple sclerosis, natalizumab (monoclonal antibody to heterodimer of α4β7 and α4β1) has been an FDA-approved treatment since 2004 [11]. The therapeutic effect of natalizumab has been shown to occur via blockade of α4β1- rather than α4β7-integrin. This is supported by the fact that antibodies against α4β7 heterodimer or β7-integrin fail to downregulate disease [12]. Furthermore, endothelial overexpression of mucosal vascular addressin cell-adhesion molecule 1 (MAdCAM-1, the counter-receptor of α4β7-integrin) does not correlate with experimental autoimmune encephalomyelitis (EAE) severity [13]. On the other hand, α4β7 is the main integrin involved in ulcerative colitis and Crohn’s disease. It is substantiated by evidence that anti-α4β7 antibodies (Vedolizumab, Millennium Pharmaceuticals) can prevent T cells binding to MAdCAM-1 in the mucosal vasculature and thus were approved by the FDA for use in ulcerative colitis and Crohn’s disease in 2014 [14,15]. The major side effects from using systemic anti-integrin therapies are progressive multifocal leukoencephalopathy (PML) and reactivation of John Cunningham (JC) virus infection, which are both serious and potentially fatal [14,16]. These have been reported with the use of natalizumab and efalizumab (from Raptiva and Genentech, respectively), a humanized monoclonal antibody to the alpha chain (CD11a) of the β2-integrin LFA-1 [14,16]. The side effect to natalizumab was proposed by targeting α4β1 [14].

The increased expression of CAMs during uveitis has been demonstrated in histopathological specimens in humans [17,18,19,20]. Blocking these pathways has been shown to be effective to treat some ocular diseases. In allergic conjunctivitis, levocabastine, an antihistamine eyedrop, has been reported to suppress inflammation via downregulating α4β1 expression at the conjunctival level [21]. Lifitegrast, a topically administered LFA-1 antagonist, has been reported to inhibit binding between T cells and ICAM-1, thus alleviating CD4^+^-T-cell-mediated dry eye symptoms and was approved for use by the FDA in 2016. It demonstrated a comparative treatment effect to topical cyclosporin A, a potent T-cell activation inhibitor [22,23]. There have been local side effects reported, such as eye irritation, dysgeusia, and reduced visual acuity. However, no systemic side effects have been observed [23], hence the rationale for anti-CAM therapy, specifically anti-integrin therapy, being used for another ocular inflammatory disease, NIU. There have been some reports using anti-integrin therapy in experimental autoimmune uveitis (EAU), an animal model used to study NIU. This review focuses on the role of anti-CAM therapy in uveitis using EAU models and provides new clinical insights.

## 2. Material and Methods

This review includes published articles on the pharmaceutical efficacy of targeting adhesion molecules for NIU and EAU, including phase 2 and phase 3 clinical trials and review papers. We performed a search on PubMed for the peer-reviewed literature available using the MeSH terms “integrin,” “α4β1,” “α4β7,” “αVβ3,” “αVβ5,” “αVβ1,” “VLA-4,” “LFA-1,” “ICAM-1,” “vascular cell adhesion protein 1 (VCAM-1),” “selectin,” “uveitis,” and/or “clinical trials.” We did not set for language limitations. The search was undertaken for published manuscripts addressing integrin-related pathophysiology in NIU, EAU, and relevant clinical trials until June 2021.

The primary search identified 339 records for the MeSH terms mentioned above and one extra record was recruited from http://clinicaltrial.gov/ (29 June 2021). A total of 254 records, which failed to meet the inclusion criteria (irrelevant to uveitis but other systemic autoimmune disease, *n* = 42) or were duplicated (*n* = 212) between the searches, were excluded (29 June 2021). Next, 85 full-text articles were assessed for eligibility, from which 67 studies were selected for qualitative synthesis and were included in the study. Two researchers (Y.-H.C., V.C.) identified the 67 published studies that met the inclusion criteria. A description of the full identification process is summarized in Figure 1.

## 3. Results

### 3.1. Role of Cell Adhesion Molecules in Experimental Autoimmune Uveitis

Differential expression of CAMs during EAU has been reported. In EAU models, the most addressed CAMs are ICAM-1 and its ligand (LFA-1; a β2-integrin). Basal retinal expression of ICAM-1 is detectable before intraocular inflammation and the level rises one to two days prior to the clinical manifestation of EAU [24]. This is supported by in vitro evidence that ICAM-1 is constantly expressed on retinal pigment epithelium (RPE), and its expression level is upregulated after inflammation is induced [25,26,27]. LFA-1 is reported to be expressed on 40% of uveitogenic CD4^+^ T cells and 60% of non-CD4^+^ T cells at the early stages of the EAU model in C57BL/6 mice (day 12), and its level is upregulated throughout the course of EAU [28,29]. Both ICAM-1 and LFA-1 inhibitors can prevent effector T cells from migrating into the retinae in vivo and reduce clinical EAU severity [29,30,31]. In vitro assays further demonstrated that anti-ICAM-1 antibody blocked non-uveitogenic, polarized Th1 and Th17 cell transmigration at similar levels across the human retinal endothelial cell (REC) monolayer in a Boyden chamber [32]. The effect of anti-LFA-1 antibody was reported to bypass CD4^+^ regulatory T cells (Tregs) [31,33]. In an ocular surface disease mouse model, it was demonstrated that an LFA-1 antagonist (Lifitegrast) downregulated clinical dry eye severity via preferentially blocking Th1 cell migration [34].

On the other hand, although VCAM-1 is expressed neither in healthy retinal tissues nor on unstimulated retinal barrier cells in vitro, its expression is induced and upregulated on RECs and RPEs during inflammation [25,26,27]. The expression of VLA-4 (α4β1 integrin), the ligand to VCAM-1, has been reported mostly in non-CD4^+^ T cells and rarely (2%) on uveitogenic CD4^+^ T cells during the early stages of EAU in C57BL/6 mice [28]. Our data demonstrated that at peak classical EAU in B10.RIII mice, there was an equal expression (30~40%) of VLA-4 by both Th1 and Th17 cells [35]. The increase in VLA-4 expression by CD4^+^ T cells was confined to the draining lymph nodes and the eye, but not the peripheral blood, suggesting that pathogenic leukocytes expressing VLA-4 are only present at local sites of inflammation [35,36]. This is supported by an in vitro study showing that antibody to VCAM-1 does not affect activated non-uveitogenic Th1 and Th17 cell migration [32]. Targeting VLA-4 is able to block the recruitment and migration of uveitogenic T cells and monocytes across the blood–retinal barrier (BRB), which ameliorates EAU development without affecting systemic immunity. A recent study reported that a small molecule inhibiting VLA-4 ameliorated the clinical signs of EAU by preferentially suppressing Th17 cell and inflammatory monocyte transmigration across the eye [35]. In addition, the effects of targeting VLA-4 and LFA-1 were reported to be transient and continuous treatment is needed [33]. It has been shown that natalizumab (less than 1.25 mg) is safe for intravitreal injection in rabbit eyes, suggesting its future potential application in uveitis. A higher dose of intravitreal natalizumab resulted in damage to the photoreceptor synaptic terminals and inner nuclear layer in the retina, although no systemic side effect was noted [37]. Intravitreal injections of integrin inhibitors may only target immune cells locally at the site of inflammation and vascular permeability breakdowns, a localized approach that may cause fewer systemic adverse side effects of concern. It may therefore be a potential therapeutic strategy for future studies.

Several in vivo EAU studies have demonstrated capability of anti-ICAM-1 Ab and anti-LFA-1 Ab to ameliorate clinical severity in different EAU animal models but had no effect on experimental melanin-induced uveitis [38,39,40,41]. These indicated that ICAM-1/LFA-1 pathways are primarily involved in Th1-dominant uveitis, but do not have a role in cancer-associated uveitis. Blocking α4 also partially suppress uveitogenic leukocyte activation, indicating the role for other CAMs [40]. In vivo studies demonstrated that the distribution of other cell adhesion molecules such as P-selectin, E-selectin, and PECAM-1 increased progressively on the retinal microvascular endothelium alongside the development of EAU [9,24]. P-selectin and E-selectin were reported to be preferentially up-regulated at the retinal venules, which are the main site of leukocyte extravasation [42]. In a model of acute anterior uveitis, endotoxin-induced uveitis (EIU), it was demonstrated that blocking P-selectin and E-selectin was able to reduce leukocyte infiltration to the eye [43,44,45]. It has also been reported that rolling and recruitment of Th1-cell transmigration across the BRB is preferentially mediated by PSGL-1:P/E-selectin interactions in EAU [42]. The results are summarized in Table 1.

The CAMs that are expressed on the leukocytes, such as VLA-4 and LFA-1, are upregulated throughout the course of EAU and if suppressed, are believed to downregulate EAU via decreasing uveitogenic cell transmigration across the BRB. The CAMs expressed at the vascular endothelium site are P-selectin, E-selectin, ICAM-1, and VCAM-1. Blocking P-selectin and E-selectin interferes with leukocyte adherence to the vascular endothelium. Hence, they are more important at the disease initiation phase than during the chronic phase. In later phases of disease, suppression of ICAM-1 and VCAM-1 becomes more important to prevent structural damage of the retina resulting from continuous leukocyte transmigration into the eye.

### 3.2. Role of Adhesion Molecules in NIU

The ICAM-1/LFA-1 and VCAM-1/VLA-4 are the main pathways reportedly involved in uveitis development in the literature. There is direct evidence of VCAM-1 as well as ICAM-1/LFA involvement in uveitis development, as an increased expression was detected in iris biopsies from patients relative to controls [46]. In NIU patients, peripheral soluble ICAM-l (sICAM-1) was reported to be a potential biomarker for disease relapse and activity but not sVCAM-1 [47,48,49,50,51,52]. However, contradictory data also reported that sICAM-1 and sVCAM-1 were not associated with active uveitis specifically related to Behçet’s disease (BD) patients, and that there was only an association of sICAM-1 with the systemic activity [53]. Nevertheless, genetic studies have indicated that ICAM-1 (rs5498) is a susceptible locus for the development of BD-associated uveitis [54]. In other uveitis etiologies, however, there was evidence of other adhesion molecule involvement. For example, in patients with sympathetic ophthalmia, the expression of VLA-4, VLA-5, VCAM-1, ICAM-1, and CD44 in the peripheral blood was significantly increased in acute inflammation compared to either the disease resolution phase or normal eyes [20]. A previous study published by Haznedaroglu revealed increased levels of both sE-selectin and sP-selectin in BD patients [55]. In BD patients presenting retinal vasculitis, however, only sE-selectin and s-ICAM-1 serum levels were significantly increased [56]. In addition, it has been reported that sL-selectin from the polymorphonuclear leukocytes (PMN) decreased significantly following BD-associated uveitis flare-ups, whereas the level of Mac-1 (α_M_β2, CDl lb/CD18) remained stable [57].

To summarize, the data regarding CAMs in human uveitis are limited, with a lot of studies having been conducted several years ago. However, there is indirect evidence that ICAM-1, VCAM-1, P-selectin, and E-selectin in their soluble forms in the peripheral blood from patients are reported to be elevated compared to controls. ICAM-1 has been reported in several uveitis entities, particularly BD and sympathetic ophthalmia. It is likely that the ICAM-1 pathway plays a fundamental role in active uveitis but in different clinical presentations of uveitis. For example, in retinal vasculitis, the involvement of additional selectin families may be critical. Future studies are required to clarify how CAMs, including integrins, participate in different clinical presentations of uveitis, for example, chorioretinitis, diffuse retinitis, vasculitis, scleritis, and panuveitis, in order to develop a more personalized therapeutic approach.

### 3.3. Clinical Trials of Anti-Adhesion Molecule Therapy for NIU

There has been only one study developed to assess the safety and efficacy of an anti-CAM-based therapy for NIU as summarized in Table 2. A case report also demonstrated a downregulation of uveitis clinical activity by natalizumab (α4β1-targeted) in treating intermediate uveitis related to multiple sclerosis at one year [58]. Another case report has illustrated that Vedolizumab (α4β7-targeted) therapy is able to clinically control uveitis associated with inflammatory bowel disease [59].

It is unfortunate that subcutaneous Efalizumab therapy increased the risk of PML, as illustrated by natalizumab (monoclonal antibody to integrin α4β7 and α4β1) in treating multiple sclerosis. On the other hand, the Efalizumab study illustrated a proof-of-concept that, as observed in EAU, inhibitor to LFA-1 is able to ameliorate the clinical severity of the disease. The route of administration of CAM inhibitors might need to be taken into consideration for future studies.

### 3.4. Efficacy of Adhesion Molecule-Based Therapy in Other Retinal Disorders

Adhesion molecule inhibitors have also been used purposefully in other retinal disorders. For example, Volociximab (M200, Ophthotech Corporation, New York, NY, USA, now Iveric Bio), a chimeric mAb to α5β1 integrin, has been applied to treat choroidal neovascularization resulting from age-related macular degeneration (AMD) [61] with no side effects in dose escalation studies when it was administered intravitreally in combination with ranibizumab for enhancing an anti-permeability effect in a phase I open-labeled trial. The trial revealed a best corrected visual acuity (BCVA) gain of 10.8 letters at nine weeks. However, the effect was not distinguishable from ranibizumab (ClinicalTrials.gov Identifier: NCT00782093) [62,63]. Risuteganib (Luminate, Allegro Ophthalmics, Capistrano, CA, USA) is a small peptide targeting multiple integrins (αVβ3, αVβ5, α5β1, and α5β3). In phase II AMD studies adopting the intravitreal administration, 48% of patients gained ≥8 Early Treatment Diabetic Retinopathy Study (ETDRS) letters. BCVA from baseline to week 28 in the risuteganib arm had no severe adverse drug reaction (ClinicalTrials.gov Identifier: NCT03626636) [64]. THR-687 (Oxurion, Leuven, Belgium) is another pan integrin receptor antagonist targeting αVβ3, αVβ5, and α5β1 [65]. In an open-label, multi-center, single dose escalation phase I study for diabetic macular edema (DME) patients who responded to prior anti-VEGF and/or corticosteroid, THR-687 intravitreal injections demonstrated a BCVA gain of 11.2 letters at week 2, and a lasting effect until month 3 with no toxicities or serious adverse events being noticed (ClinicalTrials.gov Identifier: NCT03666923) [66]. SF-0166 (SciFluor Life Science, Waltham, MA, USA) is a topical eyedrop of small molecule inhibitor of integrin αVβ3. Evidence of its biologic function was demonstrated by central retinal thickness (CRT) reduction in 38% of DME patients at day 28 of use in a phase I/II prospective clinical trial (ClinicalTrials.gov Identifier: NCT02914613) [67]. In AMD and DME patients, by targeting functional arginine–glycine–aspartic acid (RGD) integrin receptors, collagen integrin receptors, and laminin integrin receptors [68], effects were demonstrated for interfering vitreo-retinal interphase (α3β1), anti-angiogenesis (αVβ3, αVβ5, and α5β1), and inhibiting astrocyte apoptosis, which contributes to vascular leakage in DR (αvβ5) [69], and promising clinical effects could be observed in preclinical and early clinical studies.

In conclusion, inhibition via targeting CAMs intravitreally in other retinal disorders such as AMD and DME illustrated the potential to serve as primary or adjunct therapy to anti-vascular endothelial growth factor agents. Nevertheless, no serious adverse events such as PML or JC virus infection were reported in these preliminary studies. This indicates that careful local administration of CAM-based therapy for uveitis patients may be a direction for future studies.

## 4. Conclusions

It has been demonstrated in proof-of-concept studies that leukocyte-specific, CAM-targeted therapy in animal studies and clinical trials can prevent immune cell migration/infiltration and reduce clinical severity (Figure 2). However, the concerns of serious systemic side effects hinder specific use of CAM-based therapies in uveitis. From CAM-based therapies attempted in AMD and DME treatment, it is likely that local intravitreal and topical eye drop treatments may exert their biological function and optimal outcomes, thereby avoiding any systemic adverse effects. In EAU studies, topical GW559090 targeting α4β1 has been shown to reduce EAU severity scores [35]. In addition, intravitreal injection of natalizumab demonstrated no systemic toxicities in vivo [37]. These studies further confirmed that these topical or intravitreally delivered drugs entered the eye and were effective locally at inflamed areas where the endothelium integrity was impaired. Currently, there is an immense unmet clinical need for new therapies to address reducing treatment burden and suboptimal outcomes to improve our current standard of care of NIU by using corticosteroid and other immunosuppressants. CAM inhibitors reduce leukocyte migration, contributing to an anti-inflammatory milieu in the retina, and may have the potential to address the treatment burden as an adjunct therapy, which requires further study. Furthermore, the data described in this review suggest that a more personalized medicine approach should identify the predominance of Th1 or Th17 involvement, which would then predict a localized targeting of ICAM-1/LFA-1 or VCAM-1/VLA-4 pathways, respectively. Similarly, if retinal vasculitis is involved, P-selectin or E-selectin directed therapy could be considered. However, to reach this stage, a better understanding of the molecular pathways involved is required, as well as a further in-depth patient stratification.

## Figures and Tables

**Figure 1 ijms-23-00503-f001:**
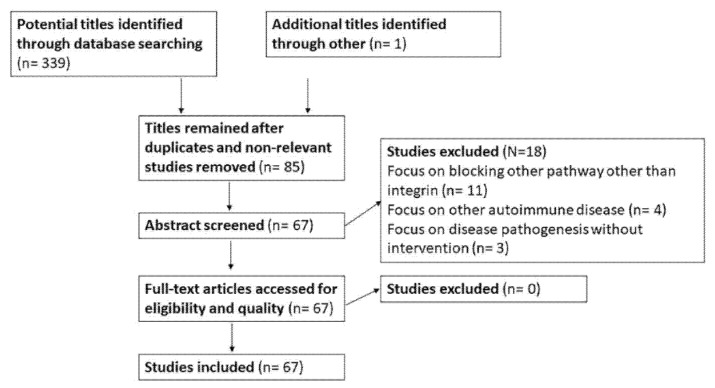
A PRISMA flow diagram depicting information through the different phases of this review.

**Figure 2 ijms-23-00503-f002:**
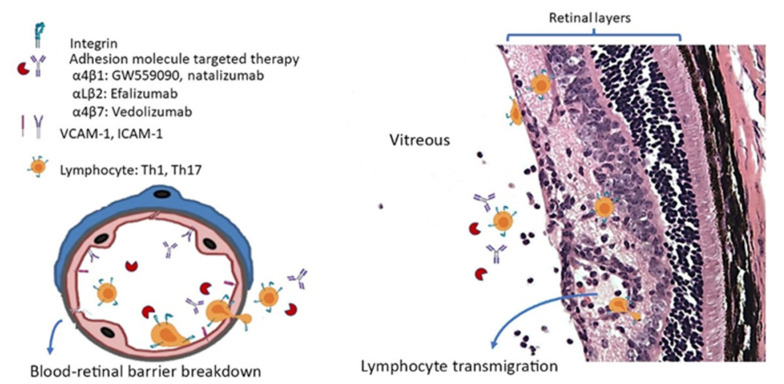
A schematic figure illustrating how adhesion molecules are involved in the process of leukocyte transmigration in uveitis and the current adhesion molecule targeted therapy being used in experimental autoimmune uveitis and non-infectious uveitis.

**Table 1 ijms-23-00503-t001:** Anti-adhesion molecule-based therapy for experimental autoimmune uveitis.

Studies	Molecule and Animal Model	Outcome
Rosenbaum et al. [38]	Ab to LFA-1 (CD11a/CD18) and ICAM-1 (iv) in rabbit model of uveitis.	Anti-CD18 Ab effectively reduced cellular infiltration if injected after 24 h of induction. Anti-CD11a Ab was effective only in the IL-1-induced model. Anti-ICAM-1 Ab was ineffective.
Uchio et al. [39]	Anti-ICAM-1 Ab or anti-LFA-1 α chain Ab (iv) consecutively after EAU induction in rat.	The development of EAU could be completely prevented by anti-ICAM-1 Ab but partially by anti-LFA-1 α chain Ab.
Martin et al. [40]	α4 active peptide inhibitor (ip) was administered to classical and adoptive transferred B10.RIII EAU mice serially at afferent and efferent phase of disease.	Treatment at afferent and efferent phase has a similar extent of disease downregulation, however; it did not ablate the disease fully. Repeated injections can reduce the disease scores further.
Smith et al. [41]	Anti-ICAM-1 Ab (ip) sequentially after induction of EMIU.	Failed to suppress leukocyte infiltration.

Ab: antibody; EMIU: experimental melanin-induced uveitis; ip: intraperitoneal injection; iv: intravenous injection.

**Table 2 ijms-23-00503-t002:** Integrin based therapy for non-infectious uveitis (NIU).

Clinical Trial	Receptor and Mechanism	Outcome
An open-label, prospective, noncomparative phase I/II clinical trial (ClinicalTrials.gov number, NCT00280826.) [60]	Weekly subcutaneous Efalizumab (a humanized form of a murine IgG1 antibody to CD11a, the α subunit of LFA-1, Raptiva; Genentech Inc., San Francisco, CA, USA) treatment for 16 weeks for patients with macular edema secondary to NIU.	Improvement in uveitis severity and macular edema. Upregulation of CD56^bright^ regulatory NK cell population in the peripheral blood. Side effects: neutropenia (17%) and headache (50%), resolved without sequelae. Efalizumab was taken off the market due to side effect of progressive multifocal leukoencephalopathy (PML).

## Data Availability

Not applicable.

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
