# Peer review of "Adhesion Molecule Targeted Therapy for Non-Infectious Uveitis"

_ijms, 2022, doi:10.3390/ijms23010503_

Round 1

Reviewer 1 Report

This is an interesting paper on adhesion molecules and their potential as targets in non-infectious uveitis that reviews the existing literature. The structure of the paper is clear and logical, and it is well organised with appropriate subheadings. The process for identifying relevant papers is well described. Language and style are fine. Cited references are appropriate and current. Conclusions are sound based on the reviewed research. My remarks are mostly concerning some parts where some clarifications could be appropriate, and some minor flaws in spelling/formatting.

Line 41, check spacing ("β3- integrins").

Line 96, broken link ("clinical.trial.gov").

Lines 115-116, proportion of uveitogenic CD4 and non-CD4 cells positive for LFA-1 in early stage of EAU is given, but is there any data on if LFA-1 is upregulated in EAU? Or is the proportion changed from control mice?

Line 120, check abbriviation on "retinal endothelial (REC)".

Line 137, "BRB" abbreviation not spelled out.

Line 158, (check throughout), inconsistent use of –ize/-ise.

Table 1, check formatting ("anti-ICAM-1" vs "Anti- ICAM-1").

Lines 165-167, is this an increase in iris VCAM-1/ICAM-1/LFA-1 in uveitis? Or are these molecules not present at all in controls?

Line 193, what is “a fare clinical outcome”? Be more specific.

Line 203, could be useful to introduce what ranibizumab is here?

Line 208, spell out "ETDRS" when first used.

Line 212, spell out "DME" when first used.

Line 224, keep capitalisation consistent ("αVβ5" vs "αvβ5").

Line 254. check spacing ("Funding:The").

Line 256, check spacing and bold font (Statement:The).

General: The abstract highlights that integrin-targeted therapies may have a potential use as a more personalised treatment (line 17), but this concept is not really touched upon in the text. How would this work in practice? What are the missing gaps in research needed? (biomarker discovery? appropriate patient stratification?).

Thanks to the authors for an interesting read and good luck with the paper.

Author Response

This is an interesting paper on adhesion molecules and their potential as targets in non-infectious uveitis that reviews the existing literature. The structure of the paper is clear and logical, and it is well organised with appropriate subheadings. The process for identifying relevant papers is well described. Language and style are fine. Cited references are appropriate and current. Conclusions are sound based on the reviewed research. My remarks are mostly concerning some parts where some clarifications could be appropriate, and some minor flaws in spelling/formatting.

Response: We thank you for your comments. The manuscript is edited as suggested (see below).

Line 41, check spacing ("β3- integrins").

Response: We thank you for your comments. The spacing is deleted.

Line 96, broken link ("clinical.trial.gov").

Response: The link has been revised to https://clinicaltrials.gov/

Lines 115-116, proportion of uveitogenic CD4 and non-CD4 cells positive for LFA-1 in early stage of EAU is given, but is there any data on if LFA-1 is upregulated in EAU? Or is the proportion changed from control mice?

Response: In reference 29, the results demonstrated that LFA-1 is upregulated from early EAU onwards. The sentence is revised into “LFA-1 is reported to be expressed on 40% of uveitogenic CD4+ T cells and 60% of non-CD4+ T cells at the early stages of the EAU model in C57BL/6 mice (day 12) and its level is upregulated throughout the course of EAU [28,29]”.

Line 120, check abbriviation on "retinal endothelial (REC)".

Response: The sentence has been reworded as “retinal endothelial cell (REC)”.

Line 137, "BRB" abbreviation not spelled out.

Response: It has been reworded as “blood-retinal barrier (BRB)”.

Line 158, (check throughout), inconsistent use of –ize/-ise.

Response: The wordings have been revised for “summarized”.

Table 1, check formatting ("anti-ICAM-1" vs "Anti- ICAM-1").

Response: It has been reformatting for Anti-ICAM-1 Ab.

Lines 165-167, is this an increase in iris VCAM-1/ICAM-1/LFA-1 in uveitis? Or are these molecules not present at all in controls?

Response: It was reported that the expressions of iris VCAM-1/ICAM-1/LFA-1 in uveitis patients were stronger than in the controls. The sentence has been reworded as “There is a direct evidence of VCAM-1 as well as ICAM-1/LFA involvement in uveitis development as an increased expression was detected in iris biopsies from patients relative to controls.”

Line 193, what is “a fare clinical outcome”? Be more specific.

Response: The sentence has been reworded as “A case report has also demonstrated a downregulation of uveitis clinical activity score by natalizumab (α4β1-targeted) in treating intermediate uveitis related to multiple sclerosis at one year.”

Line 203, could be useful to introduce what ranibizumab is here?

Response: Since the focus is on adhesion molecules in this manuscript, we did not think it relevant to introduce ranibizumab or other anti-VEGF agents here.

Line 208, spell out "ETDRS" when first used.

Response: The abbreviation has been spelled out for letters Early Treatment Diabetic Retinopathy Study (ETDRS).

Line 212, spell out "DME" when first used.

Response: The abbreviation has been spelled out for diabetic macular edema.

Line 224, keep capitalisation consistent ("αVβ5" vs "αvβ5").

Response: We have revised in text.

Line 254. check spacing ("Funding:The").

Response: It has been revised.

Line 256, check spacing and bold font (Statement:The).

Response: It has been revised.

General: The abstract highlights that integrin-targeted therapies may have a potential use as a more personalised treatment (line 17), but this concept is not really touched upon in the text. How would this work in practice? What are the missing gaps in research needed? (biomarker discovery? appropriate patient stratification?).Thanks to the authors for an interesting read and good luck with the paper.

Response: It is evident that integrins are involved during uveitis, and the concept has been validated in numerous animal studies. However, due to the paucity of cells within the eye and the technical limitations of detecting the characteristics of individual cells in the eye, there is still an unmet need as to how integrin-targeted therapies are involved. We have discussed the potential use of CAM-directed therapy and data described in this review suggest that a more personalized medicine approach should identify the predominance Th1 or Th17 involvement, which would then predict a localized targeting of ICAM-1/LFA-1 or VCAM-1/VLA-4 pathways respectively. However, to reach this stage, a further understanding of the molecular pathways involved is required as well as a further in-depth patient stratification.

Reviewer 2 Report

  1. Chen and coworkers provide a comprehensive review about “ Adhesion molecule targeted therapy for non-infectious uveitis”. Interestingly, many of the papers cited here are 20 or more years old, which indicates a. that there has not been much evolution in this field and b. deserves to be discussed on the background of potential clinical applications as postulated in the abstract, but not discussed thereafter. An additional chapter before the conclusion might provide the authors’ opinion and an outline, where the story is likely to develop, and which clinical applications might be interesting and at the horizon of the next 5 years. Further comments:
  • The term integrin and Adhesion molecule are used in parallel, but the link is not explained.
  • The sentence “The importance of integrins involved differentially in the pathways of many autoimmune diseases has been validated by its success in clinical use” is not self-explaining and should be supplemented with examples and references.
  • In- and exclusion as well as non-relevance criteria not defined à urgently needed given the fact that 255 papers have been excluded based thereof.
  • Table 1: Obviously contradictory results in the cited papers dserve to be detailed and explained to make the table understandable.
  • A summarizing statement after each of the 4 sub-chapters to chapter 3 would be appreciated by the majority of readers.
  • Chapter 3.2: Long list of informations without evident interconnection. Suggest to re-write this chapter, evtlly also 3.1 for the same reason.
  • Chapter 3.3 and table 2: add route of administration where missing…

Author Response

  1. Chen and coworkers provide a comprehensive review about “ Adhesion molecule targeted therapy for non-infectious uveitis”. Interestingly, many of the papers cited here are 20 or more years old, which indicates a. that there has not been much evolution in this field and b. deserves to be discussed on the background of potential clinical applications as postulated in the abstract, but not discussed thereafter. An additional chapter before the conclusion might provide the authors’ opinion and an outline, where the story is likely to develop, and which clinical applications might be interesting and at the horizon of the next 5 years. Further comments:

Response: An additional chapter regarding the clinical implication has been added before prior to conclusion as suggested.

  1. The term integrin and Adhesion molecule are used in parallel, but the link is not explained.

Response: Thank you for the comments. We have added the explanation to integrin and adhesion molecule to introduction (line 37-40).

  1. The sentence “The importance of integrins involved differentially in the pathways of many autoimmune diseases has been validated by its success in clinical use” is not self-explaining and should be supplemented with examples and references.

Response: Thank you’re the comments. We have explained the sentence by using adhesion molecule inhibitor in the treatment of autoimmune CNS disease, multiple sclerosis as an example in this paragraph (Line 53-72).

  1. In- and exclusion as well as non-relevance criteria not defined à urgently needed given the fact that 255 papers have been excluded based thereof.

Response: There were many adhesion molecules associated papers being included initially, however; of these, many of them were excluded because they were irrelevant to uveitis. We have clarified this in material and method section.

  1. Table 1: Obviously contradictory results in the cited papers dserve to be detailed and explained to make the table understandable.

Response: We have added explanation to the table between line 154-160.

  1. A summarizing statement after each of the 4 sub-chapters to chapter 3 would be appreciated by the majority of readers.

Response: We have added a summarizing statement at the end of each sub-chapters to chapter 3.

  1. Chapter 3.2: Long list of informations without evident interconnection. Suggest to re-write this chapter, evtlly also 3.1 for the same reason.

Response: We have re-written 3.1 and 3.2.

  1. Chapter 3.3 and table 2: add route of administration where missing…

Response: We have added route of administration.

Round 2

Reviewer 2 Report

The authors responded well to the reviewer input and provided a well-readable manuscript not needing further revision.